

# Nitrous oxide (N₂O) in the sea surface microlayer and underlying water during a phytoplankton bloom: a mesocosm study

Ina Stoltenberg[1], Lea Lange[1], and Hermann W. Bange[1]

[1]Marine Biogeochemistry, GEOMAR Helmholtz Centre for Ocean Research Kiel, Kiel, 24148, Germany

*Correspondence to*: Ina Stoltenberg (istoltenberg@geomar.de), Hermann Bange (hbange@geomar.de)

**Abstract.** Nitrous oxide ($N_2O$) is an important climate-relevant atmospheric trace gas. The open and coastal oceans are a major source for atmospheric $N_2O$. However, its production and consumption pathways in the ocean are not well-known and its emissions estimates are associated with a high degree of uncertainty. Potential $N_2O$ production pathways in the oxic surface ocean include microbial nitrification, release from phytoplankton and photochemodenitrification. In order to decipher the effect of a phytoplankton bloom on dissolved $N_2O$ concentrations, $N_2O$ was measured – for the first time – in the sea surface microlayer (SML, i.e. the upper 1 mm of the water column) and in the corresponding underlying water (ULW) during a mesocosm study with Jade Bay (southern North Sea) water from 16 May to 16 June 2023. $N_2O$ concentrations were slightly enriched in the SML compared to the ULW although the difference of the mean $N_2O$ concentrations between the ULW and SML was statistically not significant. However, the enrichment of $N_2O$ in the SML was most probably underestimated due to the loss of $N_2O$ during sampling with the glass plate method. $N_2O$ was supersaturated (100% – 157%) in the ULW and SML during the course of the study which indicated an in-situ production of $N_2O$. $N_2O$ in-situ production was most probably driven by photochemodenitrification in combination with the release from phytoplankton whereas microbial production of $N_2O$ via nitrification appeared to be of minor importance. $N_2O$ concentrations in both the ULW and the SML were remarkably constant over time and were apparently not affected by irradiation and a phytoplankton bloom which was triggered by nutrient additions. We therefore conclude that the $N_2O$ in-situ sources were balanced by the release of $N_2O$ to the atmosphere resulting in a steady state of the system. Our results indicate that the role of the SML for $N_2O$ cycling in the surface ocean and its emissions to the atmosphere has been overlooked so far. Moreover, our results are in line with results from field studies which showed that phytoplankton blooms in the ocean do not result in temporarily enhanced $N_2O$ concentrations in the ocean surface layer.

## 1 Introduction

Nitrous oxide ($N_2O$) is a climate-relevant, long-lived trace gas in the Earth's atmosphere: In the troposphere it acts as a strong greenhouse gas and in the stratosphere it is one of the major ozone-depleting compounds (IPCC, 2021). The open and coastal oceans contribute about 25 % to the natural and anthropogenic emissions of atmospheric $N_2O$ (Tian et al., 2024). Natural $N_2O$ production is part of the nitrogen cycle where it occurs as a by-product during nitrification (i.e. microbial oxidation of ammonia to nitrite and nitrate) and as an intermediate during denitrification (i.e. microbial reduction of nitrate via $N_2O$ to dinitrogen) (see e.g. Bange et al., 2024) Only recently it was shown that in aquatic environments $N_2O$ is also produced photochemically via phtotochemodenitrification from dissolved nitrite (Leon-Palmero et al., 2025). Moreover, $N_2O$ is also released from cultures of marine phytoplankton (McLeod et al., 2021; Plouviez et al., 2019; Teuma et al., 2023). However, phytoplankton blooms in the ocean




which were triggered by artificial or natural iron fertilization showed that phytoplankton blooms not necessarily lead to enhanced $N_2O$ production (Farías et al., 2015; Law and Ling, 2001; Walter et al., 2005). $N_2O$ cycling in the ocean is, thus, usually described as being dominated by microbial production and consumption pathways such as nitrification and denitrification. The contributions by its photochemical production and release by phytoplankton are unknown or associated with large uncertainties, respectively.

The sea surface microlayer (SML) forms the interface between the ocean and the atmosphere. It is ubiquitous in the open and coastal oceans and thus covers about 71% of the Earth's surface, with a thickness of up to 1 mm (Engel et al., 2017). The SML plays a key role for the exchange of momentum, heat, gases and aerosols between the ocean and the atmosphere. Despite its comparably small volume, it is a distinct water layer which differs from the underlying (i.e. subsurface) water (ULW) in its physical properties as well as its chemical and biological composition (Cunliffe et al., 2013; Engel et al., 2017; Wurl et al., 2017; 2021). Dissolved nutrients (e.g. nitrite) as well as surface-active organic compounds (surfactants) which originate from biological production can accumulate in the SML (Wurl et al., 2011; Zhou et al., 2018). There are direct and indirect hints that especially the accumulation of surfactants in the SML affects the exchange of $N_2O$ across the water/atmosphere interface (Kock et al., 2012; Mesarchaki et al., 2015). Moreover, processes in the SML seem to result in different transfer velocities for the release of $N_2O$ form the water side (evasion) and uptake of $N_2O$ from the air side (invasion) (Conrad and Seiler, 1988; Upstill-Goddard et al., 2003).

However, the determination of trace gases in the SML is difficult because of the limited access to the SML in combination with the inherent problems of gas loss with the usually applied SML sampling methods (i.e. the glass plate and related methods). To the best of our knowledge, $N_2O$ concentrations have not been determined in the SML so far.

Here we present the first time-series measurement of $N_2O$ concentrations in both the SML and the ULW during a mesocosm study in May/June 2023 (Bibi et al., 2025). The overarching objectives of our study were (1) to assess whether there is an accumulation of $N_2O$ in the SML and (2) to decipher the effect of enhanced biological production on dissolved $N_2O$ in the water column.

## 2 Methods

The mesocosm study was part of the BASS (Biogeochemical processes and Air–sea exchange in the Sea-Surface microlayer) project and took place in one of the mesocosms at the Sea Surface Facility (SURF) of the University of Oldenburg in Wilhelmshaven, Germany, between 16 May to 16 June 2023.

### 1.1 Mesocosm setup

A detailed description of the mesocosm facility and the BASS study is given in Bibi et al. (2025). The mesocosm basin was filled with water from the adjacent Jade Bay which is a shallow bay with water depths <20 m on the southern North Sea coast. The mesocosm basin was filled with particle-reduced Jade Bay water on 13 May 2023. Subsequently, fleece filtration and protein skimming were initiated under slow water circulation for three consecutive days. Small pumps were used (1) to ensure the homogeneity of the water column and reduce stratification and (2) to reduce particle settling and biofilm formation on walls and bottom of the basin. For details of the pre-treatment of the Jade Bay water and the setup of the pump array see Bibi et al. (2025). On 15 May 2023, the surface layer of the water column was skimmed with glass plates for nine hours to remove any visible organic



75   film residues and debris from the water surface. Regular sampling for dissolved $N_2O$ from the SML and the ULW
started on 16 May. Jade Bay water was replenished with 4.5 L per day to replace the water removed by the sampling
of the SML and the ULW in order to maintain a constant water volume in the basin.

Measurements during the study included various physical, biological and chemical parameters. Here we show the
water temperature, salinity, nitrate/nitrite and chlorophyll concentrations which correspond to our $N_2O$

80   concentration measurements. For details of the applied methods and instruments see the overview in Bibi et al.
(2025).

In order to trigger a phytoplankton bloom, nutrients (nitrate, phosphate and silicate) were added on 26 May, 30
May and 1 June 2023 (for details of the nutrient addition, such as the added concentrations, see Bibi et al., 2025).
The mesocosm facility has a retractable roof of transparent polycarbonate plates. The roof was open during the

85   day and the basin was exposed to day light whereas the roof was closed during the night and rain events. The
length of the days increased from 16 hours on 16 May to 17 hours on 16 June 2023 (see https://www.sunrise-and-
sunset.com/de/sun/deutschland/wilhelmshaven/2023/mai; last access on 15 September 2025).

**1.2 $N_2O$ sampling**

All samples for $N_2O$ were collected in triplicates in 20 mL amber glass vials, bubble-free and crimped air-tight

90   with butyl rubber stoppers and aluminium caps. SML samples for $N_2O$ were taken every three days alternating
either 30 minutes past sunrise or 10 hours past sunrise. The SML samples were taken with a glass plate (Cunliffe
and Wurl, 2014). The water from the glass plate was transferred to the glass vials using a wiper and a funnel.
Samples from the ULW were collected twice a day (30 minutes past sunrise and 10 hours past sunrise) on days
with no SML sampling for $N_2O$ and three times a day on days with SML sampling for $N_2O$ including one night

95   (dark) sample taken about two hours after sunset. Diurnal (24h) cycles were sampled from the UWL on 24 May,
2 June, 4 June and 8 June 2023 with a time interval of two hours. Samples from the ULW were taken by using a
Teflon® hose which was placed into the mesocosm using a lab stand to maintain a sampling depth of 0.6 m.

To stop any microbial or other biological processes which might influence $N_2O$ production or consumption in the
sample, all samples were poisoned as quick as possible (usually within one hour) after sampling by adding 0.05

mL of oversaturated aqueous solution of mercury(II) chloride ($HgCl_2$). Samples were inverted after poisoning to
ensure that the added $HgCl_2$ solution was distributed evenly throughout the sample. Samples were stored at room
temperature and in the dark until measurement in our laboratory at GEOMAR in Kiel. All samples were measured
within 21 months after the study. A comparably long storage time, however, should not affect the $N_2O$
concentrations as (Wilson et al., 2018) pointed out.

**1.3 $N_2O$ concentration measurements**

$N_2O$ concentrations were determined with the static headspace technique in combination with a gas chromatograph
(Hewlett-Packard 5890A Series II) equipped with an electron capture detector for separation of $N_2O$ from the gas
mixture and detection, respectively. We replaced 10 mL of the seawater sample by injecting helium with a gas
tight syringe. After agitation on a Vortex mixer, samples were left to equilibrate for two hours. Subsequently, a

subsample of 9 mL was taken from the headspace with a gas tight syringe and injected manually into the gas
chromatograph. Before flushing the 2 mL sample loop the sample was dried by passing through a moisture trap
filled with phosphorous pentoxide (Sicapent®, Merck Germany). A mixture of Argon (5 %) and Methane was used
as a carrier gas and gas chromatographic separation was executed at 190 °C on a packed molecular sieve column



(6ft ×1/8" SS, 5Å, mesh 80/100, Alltech GmbH, Germany). For calibration, two standard gas mixtures (working
standards) of $N_2O$ in synthetic air with dry mole fractions of 391.46 ± 9.80 ppb and 1055.99 ± 7.91 ppb $N_2O$ were
used (Deuste Gas Solution GmbH, Schömberg, Germany). The working standards have been calibrated against a
primary $N_2O$ standard gas mixture provided by the National Oceanic and Atmospheric Administration (NOAA
PMEL, Seattle, Wa, USA; for details see Wilson et al., 2018). The concentrations of dissolved $N_2O$ ($C_{N2O}$) in the
SML and ULW samples were computed with Eq. (1):


$$C_{N2O} = \beta*x'*P + (x'P/RT)*(V_{hs}/V_{wp}) \tag{1}$$

where $\beta$ is the Bunsen solubility (in nmol $L^{-1}$ $atm^{-1}$) of $N_2O$ calculated as a function of salinity and temperature at
the time of equilibration (Weiss and Price, 1980). T is the temperature at the time of equilibration. P is the
atmospheric pressure (set to 1 atm). V stands for the volume of the water phase (wp) and the headspace (hs) in
mL. R is the gas constant ($8.2057 \ 10^{-5}$ $m^3$ atm $K^{-1}$ $mol^{-1}$) and x' is the dry mole fraction of $N_2O$ in ppb (= $10^{-9}$).
The average relative measurement error of the average $N_2O$ concentrations (= mean of the triplicates) was ± 4.5%.

**1.4 $N_2O$ enrichment factors and saturations**

$N_2O$ enrichment factors ($EF_{N2O}$) are given as $(C_{N2O})_{SML}/(C_{N2O})_{ULW}$ and $N_2O$ saturations ($Sat_{N2O}$ in %) were
computed according to Eq. (2):

$$Sat_{N2O} = 100 * C_{N2O}/C_{eq} \tag{2}$$

where $C_{eq}$ is the equilibrium concentration of $N_2O$ calculated with the solubility Eq. of (Weiss and Price, 1980) by
using the water temperature and salinity at the time of sampling and the average monthly atmospheric dry mole
fraction of 337.6 ppb $N_2O$ for May/June 2023 measured at the AGAGE (Advanced Global Atmospheric Gases
Experiment) monitoring station in Mace Head at the west coast of Ireland ((Prinn et al., 2018); dataset
doi:10.3334/CDIAC/ATG.DB1001           accessed           via           https://data.ess-
dive.lbl.gov/datasets/doi:10.3334/CDIAC/ATG.DB1001 on 15 September 2025).

**1.5 Photochemical $N_2O$ production**

$N_2O$ production rates via photochemodenitrification ($PR_{pcd}$ in nmol $N_2O$ $L^{-1}$ $h^{-1}$) were estimated with Eq. (3) given
in (Leon-Palmero et al., 2025):

$$PR_{pcd} = 0.5/24 * 0.32 * \exp(0.23 * C_{NO2^-}) \tag{3}$$


where $C_{NO2^-}$ is the concentration of dissolved $NO_2^-$ in µmol $L^{-1}$ in the ULW or in the SML and the factor 0.5/24 is
the conversion factor from nmol N $L^{-1}$ $d^{-1}$ to nmol $N_2O$ $L^{-1}$ $h^{-1}$.

**1.6 $N_2O$ air-sea gas exchange**

A rough estimate of the average $N_2O$ gas exchange ($F_{ase}$ in nmol $N_2O$ $L^{-1}$ $h^{-1}$) was computed according to the
approach of (Liss and Merlivat, 1986) (Eq. 4). The approach of Liss and Merlivat (1986) was chosen because it
was derived mainly from wind/water tunnel studies which do have wind/wave features comparable to the





mesocosm study (e.g. in view of the wind fetch) and therefore seems to be more appropriate than the usually used approaches derived from open ocean studies which do have significantly different wind/wave features (see e.g. Wanninkhof, 2014).


$$F_{ase} = 0.17 * (0.01/d) * u_{10} * (C_{N2O} - C_{eq}) * (600 * Sc^{-2/3}) \qquad (4)$$

where $u_{10}$ is the average wind speed in 10 m height (= $0.9 \pm 0.6$ m s$^{-1}$), d is set to the average thickness of the SML during the study (0.001 m, see Rauch et al., 2025) or the overall water depth of the mesocosm basin (0.8 m), $C_{N2O}$

is the $N_2O$ concentration in the SML or in the ULW, 0.01 is the conversion factor from cm to m and Sc is the Schmidt number which was computed using the empirical Eq.s for the kinematic viscosity of seawater (Siedler and Peters, 1986) and the diffusion coefficient of $N_2O$ in water (Rhee, 2000). The overall water depth of the basin was applied with the assumption that the water column in the basin was well-mixed during the study. $F_{ase}$ was set to 0 when the roof of the mesocosm facility was closed.

**2 Results**

**2.1. Overview of general conditions and parameters in the mesocosm**

Chlorophyll a concentrations varied from 1.0 to 11.4 µg L$^{-1}$ and were affected by the nutrient additions which triggered a phytoplankton bloom (Bibi et al., 2025). Three phases of the bloom have been identified: 1) a pre-bloom phase from the start of the study until 26 May 2023, 2) a bloom phase from 27 May until 4 June 2023 and

3) the post-bloom phase from 5 June 2023 until the end of the study (Bibi et al., 2025). Haptophytes, specifically *Emiliania huxleyi* (*Gephyrocapsa huxleyi*), dominated the phytoplankton community, followed by diatoms, primarily *Cylindrotheca closterium* (Bibi et al., 2025). Enrichments of surfactants and dissolved organic carbon were observed after the bloom (data are shown in Bibi et al., 2025, Asmussen-Schäfer et al., 2025). Water temperatures were in the range of 16.6 and 20.3 °C until 2 June 2023 and then started to increase up to 24.3 °C on

12 June 2023. The salinity increased almost linearly during the study from 28.98 to 32.28 (Fig. 1).





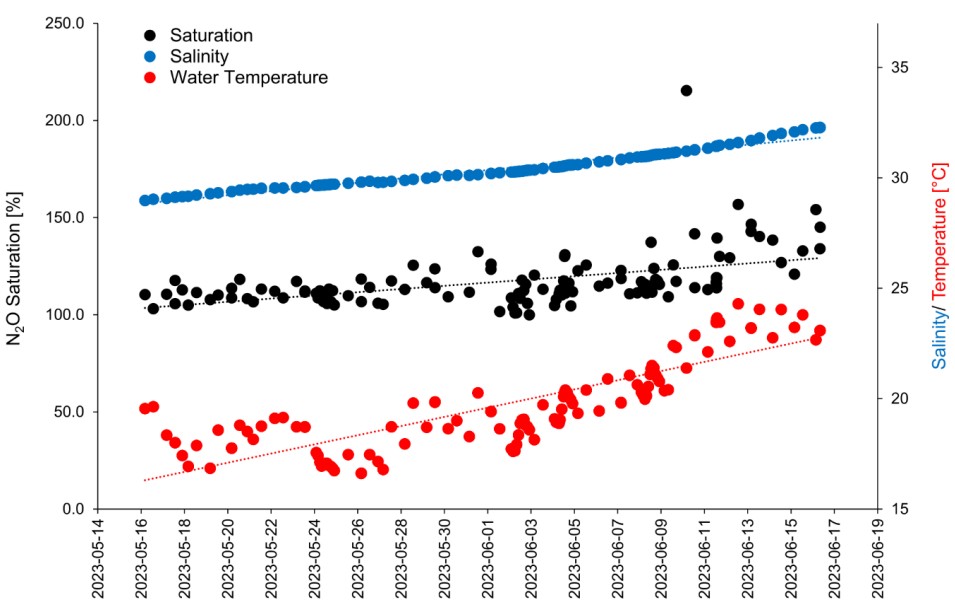

**Figure 1: Water temperature, salinity and N₂O saturation during the mesocosm study.**

Nitrate ($NO_3^-$) and nitrite ($NO_2^-$) in the ULW and the SLM, as well as nutrient additions to the mesocosm are

shown in Fig. 2 (Bibi et al., 2025).

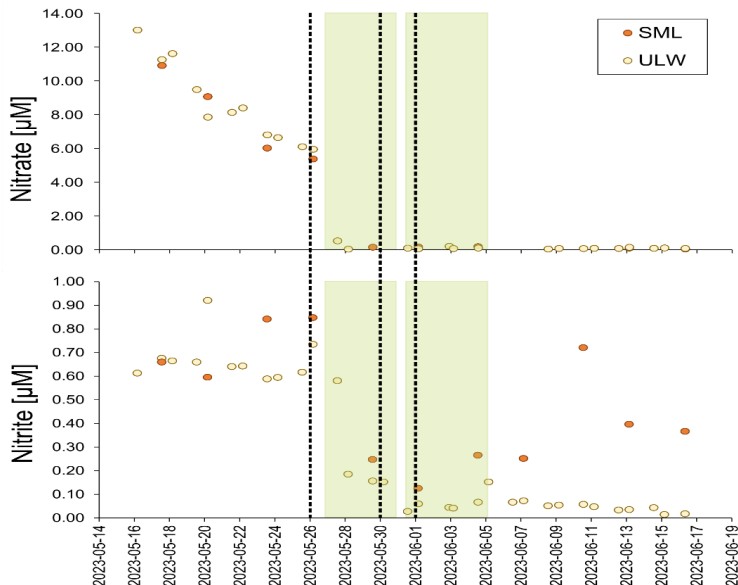

**Figure 2: Dissolved nitrate (upper panel) and dissolved nitrite (lower panel) during the mesocosm study in the SML (filled red circles) and the ULW (open circles). µM stands for $10^{-6}$ mol $L^{-1}$. The bloom is indicated by the green-shaded**
**boxes. The timing of the nutrient additions is indicated by the three dashed lines.**



NO$_3^-$ concentrations decreased steadily from the start of the study until the onset of the bloom on 27 May 2023. This was followed by a sharp drop of NO$_3^-$ concentrations which remained low (0.03 – 0.52 µmol L$^{-1}$) until the end of the study. Nitrite concentrations in both the ULW and SML dropped down as well when the bloom started on 27 May 2023. During the bloom and post-bloom phases the NO$_2^-$ concentrations remained low, whereas, the

NO$_2^-$ concentrations in the SML increased again in the post-bloom phase to a maximum concentration of >1 µmol L$^{-1}$ until 14 June 2023 (Bibi et al., 2025). With the exception of 20 May 2023, NO$_2^-$ was always enriched in the SML.

**2.2. N$_2$O concentrations, saturation and enrichment**

N$_2$O concentrations in the SML and the ULW as well as chlorophyll a concentrations are shown in Fig. 3.


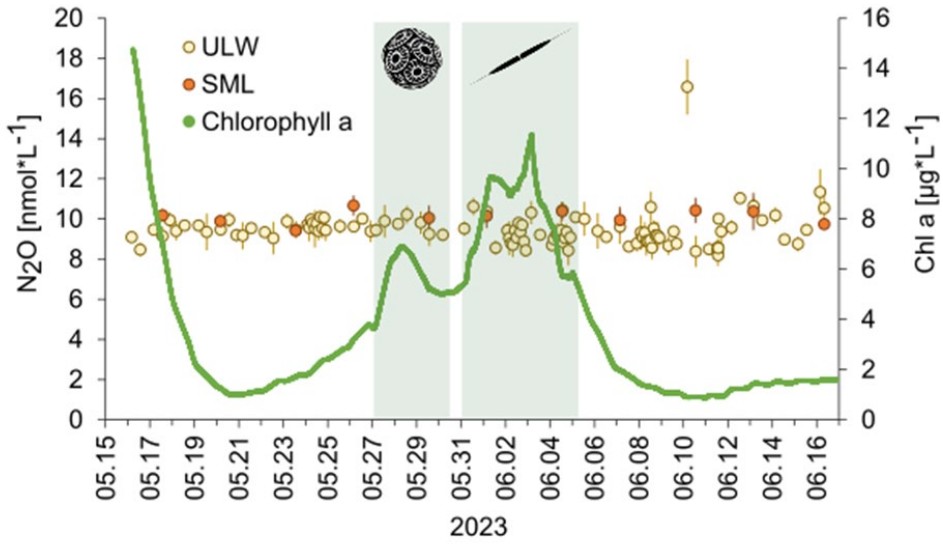

**Figure 3: N$_2$O concentrations in the SML (filled red circles) and ULW (open circles) and chlorophyll a during the mesocosm study (green line). The green shaded boxes indicate the bloom. The inserted pictures show the haptophyte *Emiliania huxleyi* (*Gephyrocapsa huxleyi*) (left) and the diatom *Cylindrotheca closterium* (right).**

N$_2$O concentrations ranged from 9.4 to 10.7 nmol L$^{-1}$ and from 8.2 to 16.6 nmol L$^{-1}$ in the ULW and the SML, respectively. There were no temporal trends for N$_2$O in the ULW and the SML. The overall average N$_2$O concentrations (± standard deviation) in the ULW was 9.4 ± 0.6 nmol L$^{-1}$ (excluding the single maximum concentration of 16.6 nmol L$^{-1}$). The average N$_2$O concentrations for the samples taken in parallel for SML and UWL were 10.1 ± 0.4 and 9.7 ± 0.7 nmol L$^{-1}$, respectively. However, the difference between the average N$_2$O

concentrations in the SML and the UWL was not significant according to the Student's t-test (two tailed, different variances, p >0.05). The average N$_2$O concentration during the day was 9.4 ± 0.7 nmol L$^{-1}$ and the average N$_2$O concentration during the night was 9.5 ± 0.6 nmol L$^{-1}$ (excluding the single maximum concentration of 16.6 nmol L$^{-1}$). Diurnal cycles of N$_2$O concentrations in the ULW and light irradiance are shown in Fig. 4. There were no diurnal trends.










**Figure 4: 24h measurements of N₂O concentrations on four days during the mesocosm study. The irradiance at a wavelength of 356 nm is shown.**

N₂O saturations in the SML and the ULW were in the range from 100 to 157 % (215 % for the single maximum
concentration from the ULW, Fig. 1). The enrichment factor of N₂O in the SML was in the range from 0.9 to 1.2
and the average enrichment factor (± standard deviation) was $1.1 \pm 0.1$ indicating an overall enrichment. However,
in a few cases $EF_{N2O} < 1.0$ was observed as well.

### 2.3. N₂O production and gas exchange

N₂O production rates via photochemodenitrification ($PR_{pcd}$) were calculated only for daylight samples and ranged
from 0.0071 to 0.0081 nmol N₂O $L^{-1}$ $h^{-1}$ and from 0.0067 to 0.0078 nmol N₂O $L^{-1}$ $h^{-1}$ in the SML and ULW,
respectively (Table 1). Although the mean $PR_{pcd}$ in the SML ($0.0075 \pm 0.0004$ nmol $L^{-1}$ $h^{-1}$) was slightly higher
compared to the $PR_{pcd}$ in the ULW ($0.0071 \pm 0.0005$ nmol $L^{-1}$ $h^{-1}$), there was no statistically significant difference
between the mean $PR_{pcd}$ in the SML and the UWL. The N₂O gas exchange ($F_{ase}$) was in the range from 0 to 4.8
nmol N₂O $L^{-1}$ $h^{-1}$ and 0 to 0.0081 nmol $L^{-1}$ $h^{-1}$ for the average thickness of the SML and the overall water depth
(including N₂O concentrations from the SML and UWl, respectively (Table 1).

Table 1: Potential N₂O sources and sinks (in nmol $L^{-1}$ $h^{-1}$). Sd stands for standard deviation.

| | Average ± sd | Minimum | Maximum | Remarks | References |
|---|---|---|---|---|---|
| Sources | | | | | |
| Photochemodenitrification in the SML | 0.0075 ± 0.0004 | 0.0071 | 0.0081 | Estimated, occurs only during day time | This study |
| Photochemodenitrification in the UWL | 0.0071 ± 0.0005 | 0.0067 | 0.0078 | Estimated, occurs only during day time | This study |
| *E. hux* (*G. hux*) | 0.11 ± 0.02 | | | culture | (McLeod et al., 2021) |
| Diatoms | | -0.01 | 0.3 | cultures | (McLeod et al., 2021, Teuma et al., 2023) |
| Nitrification | | 0.0001 | 0.0003 | Measurements from a coastal site (Boknis Eck) | (Leon-Palmero et al., 2025) |
| Sink | | | | | |
| Gas exchange (SML) | 1.7 ± 1.9 | 0 | 4.8 | Estimated for the SML (0.001 m) only; gas exchange was set to 0 when the roof was closed | This study |
| Gas exchange (SML + ULW) | 0.0014 ± 0.0018 | 0 | 0.0081 | Estimated for the overall water depth in the basin (0.8 m); gas exchange was set to 0 when the roof was closed | This study |



## 3 Discussion

### 3.1 N$_2$O concentrations in the SML

The overall average N$_2$O enrichment factor indicated an enrichment of N$_2$O in the SML. Please note, that the measurements of N$_2$O in the SML are most probably underestimated because they were not corrected for the loss of N$_2$O during sampling with the glass plate. A correction has been proven to be difficult because it depends on several, usually not quantified, factors and processes (see also Lange et al., 2025): 1) the dissolved gas saturation, because the exchange across the water/atmosphere interface on the glass plate is driven by the concentration

difference between the atmosphere and the SML water. Thus, a high supersaturation will lead to an enhanced loss of gas compared to equilibrium saturations (and vice versa), 2) the amount of the surfactants in the SML. It is well-known that increasing amounts of surfactants can reduce the N$_2$O exchange across the water/atmosphere interface (Mesarchaki et al., 2015), 3) the wind speed. When the glass plate is moved out of the water, the film of SLM water on the glass plate is exposed to enhanced wind speeds which can lead to an enhanced release of gas across

the water/atmosphere interface, and 4) the physical properties of N$_2$O, especially solubility and diffusivity, that are driven by temperature and salinity. Overall, there seems be no constant loss factor and any estimate of the loss factor, e.g. in laboratory experiments, is thus challenging because it depends mainly on in-situ field conditions (e.g. the amount of the prevailing surfactants) which are difficult to mimic in laboratory experiments. Since no estimates of the N$_2$O loss during glass plate sampling are available to our knowledge, the N$_2$O concentrations in

the SML presented here were not corrected. Based on the fact that we measured supersaturations of N$_2$O in the SML despite the occurrence of surfactants in the SML (see Bibi e al., 2025), which counteract the N$_2$O release to the atmosphere (Mesarchaki et al., 2015), we conclude that there must have been a significant enrichment of N$_2$O in the SML during the course of the mesocosm study. This is in line with the suggestion of (Leon-Palmero et al., 2025) of a UV light-driven photochemical production of N$_2$O (i.e. photochemodenitrification) which should be

more enhanced in the SML because the SML is directly exposed to the sunlight.

### 3.2 N$_2$O saturations

N$_2$O saturations in both the SML and the ULW were ≥100% during the course of the study. This supersaturation (= excess) of dissolved N$_2$O was obviously resulting from a net in-situ production of N$_2$O in the water. Therefore, the water in the mesocosm basin was a source for atmospheric N$_2$O during the course of the study. The apparent

increasing trend of the N$_2$O saturations is resulting from the increasing temporal trends of the water temperature and the salinity (Fig. 1) which resulted in a reduced N$_2$O solubility and therefore a decreasing trend in the N$_2$O equilibrium concentrations while the measured N$_2$O concentrations in the SML and the ULW showed no temporal trend (Fig. 3).

### 3.3 Sources and sink of N$_2$O

The accumulation of N$_2$O concentrations in the SML and the ULW (as reflected by the persistent supersaturations in both layers, see section above) was resulting from its in-situ production. Potential N$_2$O sources are photochemodenitrification, release from phytoplankton and microbial nitrification. The estimated photochemical production rates (PR$_{pcd}$) from both the SML and the ULW (see Table 1) are at the lower end of the so far observed N$_2$O production rates from photochemodenitrification in coastal and fresh water systems (Leon-Palmero et al.,

2025). N$_2$O production by marine phytoplankton can range from -0.06 to 0.99 nmol L$^{-1}$ h$^{-1}$ (McLeod et al., 2021).



The specific $N_2O$ production rate of *E. hux* (*G. hux*) as determined in a culture study was $0.11 \pm 0.02$ nmol $L^{-1}$ $h^{-1}$ (McLeod et al., 2021). While there is no rate available for *C. closterium* a culture study with other diatom species (incl. *Thalassiosira weissflogii, Thalassiosira pseudonana, Skeletonema marinoi* and *Cyclotella cryptica)* revealed $N_2O$ production rates from -0.01 to 0.3 nmol $N_2O$ $L^{-1}$ $h^{-1}$ (McLeod et al., 2021; Teuma et al., 2023) which are in

the same range as the rates reported for *E. hux* (*G. hux*). $N_2O$ production rates via nitrification in oxic waters, such as found in the mesocosm study with dissolved oxygen concentrations >240 µmol $L^{-1}$ (Rauch et al., 2025) are usually low: For example, $N_2O$ production rates from ammonia oxidation (i.e. the first step of microbial nitrification) measured at the Bokins Eck coastal time-series site located in Eckernförde Bay (SW Baltic Sea) were found to range from 0.0001 to 0.0003 nmol $N_2O$ $L^{-1}$ $h^{-1}$ (Leon-Palmero et al., 2025). This is in line with the fact

that prevailing high oxygen concentrations do not favour $N_2O$ production by nitrification (see e.g. Fig. 6 in Santoro et al., 2021). On the other hand, studies suggest that microenvironments around particles, including dead diatom aggregations, may provide oxygen depleted reaction space in which denitrification and $N_2O$ production may occur despite high dissolved oxygen concentrations within the surrounding water column (Ciccarese et al. 2023). However, the scale of $N_2O$ production around these particles is yet unknown. Therefore, the in-situ production of

$N_2O$ during the mesocosm study was most likely resulting from photochemodenitrification and by the release from phytoplankton with only a minor contribution from nitrification.

The time-series of $N_2O$ concentrations in both the SML and ULW showed no temporal trends which indicate that the $N_2O$ concentrations were not affected by enhanced $N_2O$ production during the bloom, especially from the two bloom-dominating species *E. hux* (*G. hux)* and *C. closterium*. On the one hand, this seems to contrast the findings

from various culture experiments which showed that haptophytes and diatoms have the potential to produce and release $N_2O$ (McLeod et al., 2021; Teuma et al., 2023). On the other hand, naturally as well as anthropogenically triggered phytoplankton blooms in the ocean did not result in enhanced $N_2O$ concentrations during the blooms (Farías et al., 2015; Law and Ling, 2001; Walter et al., 2005). The obvious negligible effect of the phytoplankton bloom on $N_2O$ concentrations during the mesocosm study (and other oceanic areas) does not exclude a $N_2O$ release

from phytoplankton per-se, but suggests that $N_2O$ release was low and that the $N_2O$ production rates resulting from culture studies may not be representative for natural ecosystems.

The high-resolution 24h-sampling for $N_2O$ in the ULW took place during the pre-bloom phase (24 May 2023), during the bloom (2 June and 4 June 2023) and during the post-bloom phase (8 June 2023). Neither the different phases of the bloom nor the solar irradiation affected the $N_2O$ concentrations (Fig. 4). This finding, in combination

with the missing trend over the entire period of the study, indicates that the $N_2O$ concentrations in both the SML and UWL were in a steady state where the in-situ sources were counterbalanced by the release of $N_2O$ to the overlying atmosphere: Rough estimates of the average $N_2O$ gas exchange in this study range from 0 to 6.5 nmol $N_2O$ $L^{-1}$ $h^{-1}$ (Table 1) and are, therefore, high enough to counteract the $N_2O$ production. Please note, however, that the approach of (Liss and Merlivat, 1986) does not account for the effect of surfactants. So, $F_{ase}$ is most probably

overestimated. However, given the high uncertainties associated with the estimates of the sources and sink of $N_2O$ listed in Table 1, we can assume that the $N_2O$ sources were balanced by the $N_2O$ gas exchange flux. The photochemical production as well as the release by phytoplankton occurs during day time only because they are light-dependent. Because the roof of the mesocosm facility was closed during the night, the wind-driven $N_2O$ gas exchange took place only during day time as well. This might explain the absence of the diurnal cycles since both

the sources and the sink of $N_2O$ were only active during the day but not during the night. This could have led to the establishment of a steady state during the day time which persisted during night time because sources and the



## 4 Conclusions

N$_2$O was measured during a mesocosm study in the ULW and, for the first time, in the SML. N$_2$O concentrations were slightly enriched in the SML, although the difference of the mean N$_2$O concentrations between the SML and

ULW was statistically not significant. However, the enrichment of N$_2$O in the SML was most probably underestimated due to the loss of N$_2$O during sampling with the glass plate method. Consequently, a significant enrichment of N$_2$O in the SML results in an enhanced N$_2$O release to the atmosphere. Therefore, estimates of N$_2$O emissions which do not account for the N$_2$O enrichment in the SML most probably underestimate the N$_2$O flux to the atmosphere. N$_2$O was supersaturated in the SML and UWL during the course the mesocosm study which

indicated an in-situ production of N$_2$O. N$_2$O in-situ production was most probably dominated by photochemodenitrification in combination with the release from phytoplankton. Microbial production of N$_2$O via nitrification was assumed to be of minor importance because of the prevailing high oxygen concentrations which do not favour N$_2$O production via nitrification. The N$_2$O in-situ sources were obviously balanced by the release of N$_2$O to the overlying atmosphere and thus the system was in a steady state. Therefore, N$_2$O concentrations in both

the SML and the UWL were remarkably constant over time showing no diurnal cycles and no enhanced N$_2$O concentrations during the phytoplankton bloom. Our results are thus in line with the results from field studies which showed that phytoplankton blooms in the ocean do not result in temporarily enhanced N$_2$O concentrations in the ocean surface layer.

The results presented here indicate that the role of the SML for N$_2$O cycling in the surface ocean has been

overlooked so far. It seems to be more important than previously thought. Future studies should, therefore, identify and quantify the N$_2$O sources and sinks in the SML. Moreover, we plea to develop a sampling method which minimizes the loss of dissolved trace gases while sampling in order to get reliable N$_2$O measurements from the SML to be used in estimates of the N$_2$O emissions to the atmosphere.

## Data availability

All data will be archived and made available to the scientific community via the PANGAEA database upon doi assignment. Data are available at any time from the authors upon request.

## Author contribution

IS: conceptualization, formal analysis, investigation, supervision, visualisation, writing (original draft preparation), writing (review and editing). LL: conceptualization, formal analysis, investigation, writing (review

and editing). HWB: conceptualization, formal analysis, funding acquisition, supervision, writing (review and editing).

## Competing interests

HWB is a member of the editorial board of Biogeosciences.



**Acknowledgements**

We would like to thank Hendrik Feil and all other colleagues of the BASS project for their role in the planning, set up and maintenance of the mesocosm study as well as during the sampling process. The authors would also like to thank the student assistants of our laboratory at GEOMAR Helmholtz Centre for Ocean Research Kiel, namely Isabell Hentschel, Laura Biet, Jonas Blendl, Romy Kreyenhagen, Daniel Brüggemann and Florian Schreiber, for their dedicated work in analysing the samples. We thank Oliver Wurl and Riaz Bibi for coordinating

the mesocosm study.

**Financial support**

This study was funded by the DFG Research Unit BASS (Biogeochemical processes and Air–sea exchange in the Sea-Surface microlayer) with grant no. 451574234.

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
