# Peer review of "Nitrous oxide $(N_2O)$ in the sea surface microlayer and underlying water during a phytoplankton bloom: a mesocosm study"

_EGUsphere, 2025_

## Author Comment (AC1)

General answer Stoltenberg et al.:

We thank the referees for their constructive and insightful comments. We have carefully addressed all suggestions and revised the manuscript accordingly, which has improved its clarity, transparency, and overall quality. Key revisions include clarifying the methodological framework and assumptions underlying the $N_2O$ flux and photochemodenitrification estimates, expanding essential methodological descriptions for context, and refining the interpretation and wording of the results, to better acknowledge uncertainties and limitations. We believe that these changes have substantially strengthened the manuscript.

Reviewer 1:

This study quantifies nitrous oxide (N2O) concentrations in the sea surface microlayer (SML, the upper 1 mm of the water column) and the underlying water (ULW) during a phytoplankton bloom in a mesocosm experiment. The authors also estimate N2O fluxes and discuss potential pathways for N2O production in the SML, including microbial nitrification, release from phytoplankton, and photochemodenitrification. Overall, this work addresses an important and understudied topic. It provides valuable data, as no previous research has measured N2O concentrations in the SML. Understanding this layer is critical because it may play a key role in regulating fluxes of this potent greenhouse gas.

My primary concern is the statistical analysis, which currently appears insufficient to fully address the research questions. A more rigorous examination of the dataset is necessary to reveal potential dynamics in N2O concentrations. The manuscript states that there are no temporal trends in N2O within the ULW and SML (line 201), that mean concentrations in both layers do not differ significantly, and that no diurnal patterns were detected (lines 201–210). However, beyond a t-test, the methods used to assess these trends are not described in detail. A simple comparison of averages is not adequate to rule out differences or identify underlying patterns. I strongly recommend that the authors apply more robust statistical approaches to explore these relationships and potential drivers among all measured variables. For example, Figure 1 shows notable changes in temperature and salinity during the experiment, could these influence N2O concentrations? Similarly, what about chlorophyll a or other parameters, such as surfactants? That might indicate the potential role of phytoplankton.

Temporal trends (including diel trends) may not be visually apparent but could emerge when covariates are incorporated into the analysis. Besides, "samples to measure N2O concentration at the SML were taken every three days alternating either 30 minutes past sunrise or 10 hours past sunrise.". Whether the sample was taken 30 minutes past sunrise or 10 hours past sunrise could be relevant for the temporal trend. Strengthening the statistical framework would significantly enhance the manuscript's contribution and provide deeper insights into the processes governing N2O dynamics at the sea surface.

We thank the reviewer for this thoughtful and constructive comment and for highlighting the importance of a rigorous statistical evaluation of potential drivers of $N_2O$ variability. We agree in principle that multivariate and more advanced statistical approaches can be valuable tools to disentangle complex relationships among physical, chemical, and biological parameters.

In the present study, however, our decision to limit the statistical analysis was based on the characteristics of the dataset and the observed behaviour of N₂O concentrations. As shown clearly in Figure 3 and the associated time series, N₂O concentrations in both the ULW and SML remain remarkably constant throughout the entire experimental period and show no significant difference over time (we now added information on the sampling time in the figure caption). These constant N2O concentrations persists despite pronounced temporal variability in other measured parameters, including light availability, temperature, salinity, chlorophyll *a*, and the progression of the phytoplankton bloom. No significant or systematic changes in N₂O concentrations are apparent with diel cycling, increasing salinity, or biological development, nor do the data suggest layer-specific differences. To further proof this we now applied a generalized additive mixed model (GAMM) to the N₂O concentration data and we also added a statistics part into the method section (for more details on the application and outcome see methods line 187 to 201 and results lines 245 to 260).

Given the absence of visible trends, gradients, or co-variability in the N₂O time series, we consider that the application of more complex statistical methods (e.g., correlation analyses or multivariate models) would not provide additional mechanistic insight beyond what is already evident from the data. In particular, statistical relationships would be difficult to interpret meaningfully in the absence of detectable N₂O variability and could give a misleading impression of underlying controls where none are supported by the observations.

To clarify this rationale, we have revised the manuscript to more explicitly state that the lack of observed temporal, diel, or vertical variability in N₂O—despite substantial changes in environmental and biological parameters—is a key result of this study, and that this motivated our conservative statistical approach. We also now better justify why simple comparative statistics were deemed sufficient in this specific context (line 240 ff).

We appreciate the reviewer's suggestion and hope that this clarification adequately explains our reasoning.

My second concern relates to the Methods section. While I understand that many details of the incubation setup are described in Bibi et al. (2025), the current manuscript still lacks essential information needed to fully understand the experimental design. Readers should not have to rely entirely on another source to grasp the methodology. For example, the depth at which ULW samples were collected and the sampling procedure should be clearly stated. A brief explanation of the glass plate method for SML sampling may be beneficial, too. A brief description of the mesocosm facility is also necessary at the beginning, including the total volume of the setup, and if the setup has a mixing system.

In line 82, the authors mention that nutrients were added to trigger a phytoplankton bloom; the exact amounts or concentrations should be provided here rather than referring to the previous article. Additionally, the manuscript notes that "Jade Bay water was replenished with 4.5 L per day to replace the water removed by sampling." The potential impact of this replenishment on N₂O concentrations should be discussed, as it could influence the interpretation of the results. I wonder if the addition of water could cause mesocosm mixing or if it may add some N₂O or dilute it.

We thank the reviewer for this comment and for emphasizing the importance of clearly describing the experimental design. As stated the mesocosm experiment was conducted within the framework of the DFG-funded BASS project and involved multiple research groups

addressing different aspects of the sea surface microlayer. To avoid extensive repetition of identical methodological descriptions across the resulting manuscripts, a dedicated overview paper (Bibi et al., 2025) was written, which provides a comprehensive and detailed description of e.g. the mesocosm setup, experimental timeline, nutrient additions, salinity development, and bloom dynamics. This overview paper is published in the same special issue to which we submitted this manuscript and to which all other manuscripts with further results were submitted.

In the present manuscript, we therefore deliberately focus on the aspects of the experimental design that are directly relevant for the interpretation of the $N_2O$ measurements, while referring to Bibi et al. (2025) for full technical and procedural details. Nevertheless, we have ensured that the key elements necessary to follow the study and its conclusions are summarized in the Methods section (e.g. the mixing system is described in line 71, sampling depth of ULW is mentioned in line 96 and further details concerning the sampling procedures are stated in paragraph 1.2 N2O Sampling). However, we have now carefully re-checked the manuscript and clarified several points to make these descriptions more explicit. E.g. concerning the daily water addition from Jade Bay we now added the total volume of the mesocosm basin (13,600 L) in line 68 ff., to highlight that the 4.5 L of daily Jade Bay water correspond to only 0.033% of the original volume and therefore are not considered to have any effect on the $N_2O$ concentrations nor on any other parameter in the basin.

We hope this approach balances clarity and conciseness and allows readers to access additional methodological detail where needed without unnecessary duplication.

Since the estimated rates of photochemodenitrification are derived from nitrite concentrations, the authors should provide a description of the analytical method used to measure nitrite, including its detection limit and sensitivity. Additionally, it would be important to discuss whether the nitrite detection limit could constrain the ability to identify diel variations in photochemodenitrification rates.

We thank the reviewer for this suggestion. While the nitrite measurements were conducted by another project partner and are described in detail in the BASS overview paper (Bibi et al., 2025), we agree that a brief summary of the analytical approach and detection limits is useful for context. We have therefore added a concise description of nitrite sampling, analysis, and detection limits to the Methods section. We now also briefly discussed the detection limit of nitrite and its relevance for the identification of diel variations in lines 329 ff.

SML sampling method: The use of the glass plate technique for collecting $N_2O$ samples from the sea surface microlayer is not ideal, as it may underestimate $N_2O$ concentration. However, the discussion provided in Section 3.1 is valuable and helps address these concerns.

We thank the reviewer for this positive feedback.

**Some minor comments below:**

Please check the section numbering.

The section numbering was revised and should now be correct.

Figures may need some work. See the asterisks on the axis titles, carefully check figure captions, and the dimensioning issue in Fig 2. It might be a problem during the preprint editing, but in my version of the manuscript, I see ULW samples as yellow circles rather than open circles.

Thank you for highlighting the issues. We carefully checked all the figures and figure captions again. The dimensioning issue in figure 2 is now resolved, the asterisks are removed and the caption of figure 2 was corrected.

The detailed explanation provided in sections 1.3 – 1.6, which includes equations that are often omitted in manuscripts, is excellent. The community will appreciate that the authors included this information.

We appreciate the kind feedback.

Did the authors check for outliers in the dataset? If the maximum concentration of 16.6 nM is an outlier, this should be explicitly stated rather than repeatedly highlighted throughout the manuscript (e.g., lines 202, 207, …).

Thank the reviewer for the suggestion, the part was now rephrased to clarify state that we excluded the data point as an outlier (Line 242).

I recommend that the authors verify the N2O gas exchange calculations and, if possible, compare the approach used (based on Liss and Merlivat, 1986) with alternative parameterizations. The estimated flux values appear unusually high (up to 4.8 nmol N2O $L^{-1}$ $h^{-1}$).

We thank the reviewer for this comment. The $N_2O$ gas exchange calculations were carefully checked and are based on the parameterization of Liss and Merlivat (1986), as described in the Methods section. This approach was deliberately chosen because it is largely derived from wind–water tunnel experiments and thus better reflects the limited fetch and small-scale wind and wave conditions of the mesocosm setup. In contrast, commonly used open-ocean parameterizations are based on conditions that are not applicable to land-based mesocosms and would therefore not be appropriate here. We have now clarified this rationale and the interpretation of the comparatively high flux values in the discussion.

Table 1. I suggest authors specify if the values from McLeod et al 2021 were cultures exposed to natural sunlight or if they were UV irradiated.

We appreciate the suggestion and added the information into the table and the text (line 281).

Note that lines 307 and 308 are missing in the preprint pdf.

We apologize for the missing part and completed the sentence as follows: "This could have led to the establishment of a steady state during the day time which persisted during night time because sources and the sink were not active during the night."

---

## Author Comment (AC2)

General answer Stoltenberg et al.:

We thank the referees for their constructive and insightful comments. We have carefully addressed all suggestions and revised the manuscript accordingly, which has improved its clarity, transparency, and overall quality. Key revisions include clarifying the methodological framework and assumptions underlying the $N_2O$ flux and photochemodenitrification estimates, expanding essential methodological descriptions for context, and refining the interpretation and wording of the results, to better acknowledge uncertainties and limitations. We believe that these changes have substantially strengthened the manuscript.

Reviewer 2

Stoltenberg et al detail measurements of nitrous oxide (N2O) in the sea surface microlayer (SML) and underlying water (ULW) during a phytoplankton bloom in a mesocosm study using coastal North Sea waters. The results show no significant SML enrichment in the SML relative to the ULW - although as the authors note the technique used for measuring this is not optimal – and also increasing N2O supersaturation in both SML and ULW over the course of a 2-month long experiment. The interpretation of the data is somewhat cursory, with limited interpretation that relies on the unproven assumption that N2O is lost from the glass plate during SML measurement, and the identification of N2O source leans heavily on other studies with only limited analysis of ancillary data. I recommend the addition of a) more details on the mesocosm set up and operations, b) further investigation/discussion re glass plate use for dissolved gas measurement in the SML and c) further analysis of the light, temperature and nitrite data to determine whether these support the interpretation of the N2O source (or not).

1. Does the paper address relevant scientific questions within the scope of BG?

Yes

1. Does the paper present novel concepts, ideas, tools, or data?

Measuring N2O in the SSM is novel

1. Are substantial conclusions reached?

No

1. Are the scientific methods and assumptions valid and clearly outlined?

No, more methodological details are required for the mesocosm; it is insufficient to only refer to another publication. For example, "fleece filtration and protein skimming" are mentioned, but there are no further details and no references. Conversely, there is repetition in the limited details provided on mesocosm set up, and also in 1.3 N2O sampling.

We thank the reviewer for the suggestions. As replied to reviewer 1, we wanted to avoid extensive repetition of identical methodological descriptions across the manuscripts with results from this extensive multidisciplinary study. To this end, a dedicated overview paper (Bibi et al., 2025) was written, which provides a comprehensive and detailed description of e.g. the mesocosm setup, experimental timeline, nutrient additions, salinity development, and

bloom dynamics. This publication is extensively cited throughout our manuscript. However, we now carefully revised the manuscript and added more information were needed (see comments to Reviewer 1), trying to avoid unnecessary repetition.

1. Are the results sufficient to support the interpretations and conclusions?

No, data for ancilliary parameters are not used sufficiently in the interpretation of the source of N2O

See answers below.

1. Is the description of experiments and calculations sufficiently complete and precise to allow their reproduction by fellow scientists (traceability of results)?

No, see comments re limited information provided on mesocosm set up

More information on the mesocosm set up is now added (see comments to Reviewer 1 and comments before).

1. Do the authors give proper credit to related work and clearly indicate their own new/original contribution?

Yes

1. Does the title clearly reflect the contents of the paper?

Yes

1. Does the abstract provide a concise and complete summary?

Generally yes, but the abstract contains statements that are not supported by the data (see comments below)

1. Is the overall presentation well structured and clear?

Yes

1. Is the language fluent and precise?

Some typos and repetition. Final sentence in Discussion incomplete

We apologize for the missing information, there must have been a cut-off during our own editing process. The sentence is nor complete again and reads as follows: "This could have led to the establishment of a steady state during the day time which persisted during night time because sources and the sink were not active during the night."
"

1. Are mathematical formulae, symbols, abbreviations, and units correctly defined and used?

Yes

1. Should any parts of the paper (text, formulae, figures, tables) be clarified, reduced, combined, or eliminated?

See comments below

1. Are the number and quality of references appropriate?

References lacking in the discussion of uncertainties associated with the glass plate method

See replies to comments below.

1. Is the amount and quality of supplementary material appropriate?

None provided

**Comments**

**Major**

Abstract

Line 16 - "H*owever, the enrichment of N2O in the SML was most probably underestimated due to the loss of N2O during sampling with the glass plate method*". This may be the case  - and the authors would have been aware of this from the outset and could have considered alternative SML sampling techniques  - but no evidence is presented in the paper so it should be removed from the abstract or rephrased to say the effectiveness of the glass sampling was not tested.

We thank the reviewer for the suggestions and deleted the sentence from the abstract. Furthermore, we rephrased the according parts in the manuscript and we now also make clear that though we are aware of the strong imitations of the glass plate when sampling trace gases, there is to date no sufficient alternative for sampling the SML for gases (e.g. lines 301 to 303).

*"was most probably driven by photochemodenitrification*". This is also speculative in the absence of supporting evidence and also potential evidence to the contrary (see below). This would benefit from further analysis if this sentence is to remain in the abstract.

We thank the reviewer for this comment. We agree that the original wording in the abstract may have appeared too bold given that photochemodenitrification was not directly measured but quantified using an approach suggested by León-Palmero et al. (2025). We have therefore revised the wording in the abstract to more clearly reflect the interpretative nature of this result. In addition, we have now further elaborated the process-of-elimination approach in the discussion section, which made us conclude that the photochemodenitrification rates were the most probable $N_2O$ source under the study conditions. While the reviewer refers to potential evidence to the contrary ("see below"), we did not identify a specific comment "below" to which this refers to. Nevertheless, we have improved the clarity throughout the manuscript to address related points raised in all subsequent comments.

*"Our results indicate that the role of the SML for N2O cycling in the surface ocean and its emissions to the atmosphere has been overlooked so far".* Although N2O has not

been previously measured in the SML, the results and interpretation do not show significant N2O cycling in the SML so this sentence overstates their importance.

We appreciate addressing this issue and we have toned down the sentence accordingly. (line 23 to 24)

Methods

Line 135-139. There is an assumption that the overlying air in the mesocosm facility reflects that measured at the clean air site at Mace Head on the west coast of Ireland, whereas Jade Bay is surrounded by urban and industrial development. Is there any published evidence that local atmospheric N2O is consistent with that measured at Mace Head? It would have been better if N2O was measured in local air during the experiment. As the mesocosm roof was closed at night (and during rain events) this could also have influenced N2O in the overlying air. As the atmospheric N2O value is used to the saturation further justification is required here.

We appreciate this suggestion, however $N_2O$ emissions from urban and industrial sources are usually not affecting in-situ atmospheric mole fractions of $N_2O$, since $N_2O$ is not reactive in the atmosphere and thus long-lived resulting in a constant high $N_2O$ atmospheric mole fraction background. Therefore, we assume the data from Mace Head to be representative for our mesocosm site. The closing of the roof was not 'airtight'. Therefore, the closing of the roof did not affect the atmospheric $N_2O$ mole fractions during night and during rain events.

Line 160. "*Fase was set to 0 when the roof of the mesocosm facility was closed*". The water was still being circulated by pumps so there would still have been some exchange particularly as the water supersaturated relative to the overlying air.

We agree, however since the roof was closed the exchange of $N_2O$ across the water/atmosphere interface was solely driven by diffusion on the water side, which was additionally hampered by the presence of surfactants. This results in an extremely low air sea exchange because of the missing wind which justifies our assumption of $F_{ase}$ of 0 when the roof was closed.

Results

Line 190 & Figure 2. It should be noted that nitrite is higher in the SML than the ULW throughout most of the experiment. This seems to counter the interpretation that photochemodenitrification is the source of the N2O? This should be discussed.

The nitrite concentrations are majorly driven by (micro)biological processes, while the photochemodenitrification is only of minor importance for the overall nitrite concentrations and accumulation. We now added a sentence to the discussion part for clarification.

Lines 214-218. There appears to be a difference in N2O saturation **between experiment phases** with greater deviation between the SML and ULW in Phase 3. This should be mentioned and considered in the Discussion in relation to bloom phase and phytoplankton source of N2O

We thank the reviewer for this observation. While a visual inspection of the data may suggest a larger deviation between SML and ULW N₂O saturation during Phase 3, this pattern is not supported by statistical analysis. Paired comparisons between SML and ULW conducted separately for each sampling event using either paired t-tests or Wilcoxon signed-rank tests (depending on data distribution) revealed no statistically significant differences at any time point throughout the entire experiment (all p > 0.05).

Furthermore, the direction and magnitude of the observed SML–ULW differences were not consistent across sampling dates, including during Phase 3, indicating that the apparent deviations are likely driven by short-term variability rather than a systematic phase-dependent effect. To clarify this point, we have now included the full table of statistical test results in the Results section.

Given the absence of statistically supported differences, we refrained from further interpretation of phase-specific SML–ULW deviations in the Discussion, as attributing these visually perceived patterns to bloom phase or phytoplankton-driven N₂O production would be speculative.

| Date/Time | test type | p value | statistic | df | conf low | conf high | mean diff | SML mean | ULW mean |
|---|---|---|---|---|---|---|---|---|---|
| 17.05.2023 13:33 | Wilcoxon | 0.0809 | 9 | NA | NA | NA | 1.04 | 10.2 | 9.14 |
| 20.05.2023 04:09 | t-test | 0.254 | 1.56 | 2.08 | -1.92 | 4.23 | 1.16 | 9.89 | 9.47 |
| 23.05.2023 13:16 | t-test | 0.844 | -0.219 | 2.44 | -1.02 | 0.908 | -0.0583 | 9.42 | 9.47 |
| 26.05.2023 03:44 | t-test | 0.0576 | 2.95 | 3.1 | -0.0623 | 2.16 | 1.05 | 10.7 | 9.62 |
| 29.05.2023 13:26 | Wilcoxon | 0.19 | 8 | NA | NA | NA | 1.46 | 10.1 | 9.25 |
| 01.06.2023 03:33 | Wilcoxon | 0.383 | 2 | NA | NA | NA | -0.217 | 10.2 | 10.4 |
| 04.06.2023 13:04 | t-test | 0.881 | -0.16 | 3.95 | -1.06 | 0.95 | -0.0576 | 10.4 | 10.4 |
| 07.06.2023 03:30 | t-test | 0.615 | 0.547 | 3.77 | -1.4 | 2.06 | 0.333 | 9.94 | 9.61 |
| 10.06.2023 13:16 | Wilcoxon | 0.149 | 6 | NA | NA | NA | 2.05 | 10.4 | 8.38 |
| 13.06.2023 03:37 | t-test | 0.661 | -0.491 | 2.67 | -2.19 | 1.64 | -0.275 | 10.4 | 10.7 |
| 16.06.2023 08:00 | t-test | 0.288 | -1.43 | 2 | -3.22 | 1.61 | -0.806 | 9.74 | 10.5 |

Line 219. As photochemodenitification rate is calculated using nitrite concentration, the change in the nitrite concentration between experiment days and phases, and between night and day, should be considered.

We thank the reviewer for this comment. Nitrite concentrations used to calculate photochemodenitrification rates were measured during daytime only, with samples collected in the morning and afternoon. No nitrite measurements were available for night time periods. Consequently, potential diel variability in nitrite concentrations could not be explicitly resolved. The photochemodenitrification rates were therefore calculated using the available daytime nitrite concentrations, consistent with the fact that the process itself is light-dependent and expected to be absent during darkness. Temporal changes in nitrite concentrations between experimental phases and days are reflected in the rate calculations through the exponential dependence on $C_{NO_2^-}$. However, we acknowledge that periods during which nitrite concentrations fell below the analytical detection limit (0.4 µmol L⁻¹) introduce additional uncertainty. We have clarified these limitations in the revised manuscript.

Discussion

Line 230. "The overall average N2O enrichment factor indicated an enrichment of N2O in the SML." Clarify here that this enrichment is not statistically significant

We appreciate the suggestion and rephrased this sentence accordingly (line 285).

Lines 235. The surfactant concentration also influences the amount of water retained on the glass plate, and so arguably the N2O concentration.

We agree that surfactant concentrations influence the amount of water retained on the glass plate, however the initial original concentration of $N_2O$ on the glass plate is not influenced or changing by the retained water volume on the glass plate.

Lines 239. "the film of SLM water on the glass plate is exposed to enhanced wind speeds". Its not so much a different windspeed, as the angle of exposure to the air movement and shear are different.

We appreciate the note and changed the text accordingly by changing wind speed to altered shear condition (line 295).

Line 242 "Overall, there seems be no constant loss factor …" Its unclear what this sentence is referring to, as there have been no previous studies of N2O in the SML but also there is no analysis of loss in this paper (other than qualitatively in the previous sentences).

We understand why this wording is misleading and we rewrote the sentence to clarify the statement (line 299). What was meant was a universal loss factor which applies to and considers any condition in the field. Which is obviously impossible and therefore there is no general loss factor that could be properly used to correct SML measurements.

Line 245. "we measured supersaturations of N2O in the SML despite the occurrence of surfactants in the SML which counteract the N2O release…". Why "despite"? Wouldn't suppression of gas exchange by surfactants enhance N2O supersaturation in the SML?

We agree the statement was confusing, we deleted the middle part "despite the occurrence of surfactants in the SML (see Bibi e al., 2025), which counteract the N2O release to the atmosphere (Mesarchaki et al., 2015)" to clarify the statement.

Line 247. "we conclude that there must have been a significant enrichment of N2O in the SML during the course of the mesocosm study." Its not clear what this conclusion is based on other than assumption of loss from the glass plate. There are studies for other gases that have compared recovery of the glass plate with other SML techniques (see references below).

We now refined the wording in the section above this line to further clarify our conclusion and we added the suggested references (line 306 ff.).

Line 248. "This is in line with the suggestion of a UV light-driven photochemical production of N2O (i.e. photochemodenitrification) which should be more enhanced in the SML because the SML is directly exposed to the sunlight."

Line 279 "the in-situ production of N2O during the mesocosm study was most likely resulting from photochemodenitrification",

Line 302 that "photochemical production…. occurs during day time only because they are light-dependent".

Contrary to these statements (above) there doesn't appear to be any evidence of a or nitrite concentration with experiment day or day-night variation in light in the SML? Further analysis comparing N2O, nitrite and light availability is required to support this interpretation and conclusion that photochemodenitrification is the source.

The diurnal cycles which we described in the manuscript are not based on SML samples but on ULW samples. The conclusion of a photochemodenitrification driven source is based on the estimated rates of $N_2O$ production given in Table 2 (former Table 1). The missing trend in $N_2O$ concentrations over the whole experiment but also between day and night is presumably due to balancing losses and consumption as described in line 379 to 381 "This could have led to the establishment of a steady state during the day time which persisted during night time because sources and the sink were not active during the night."

Line 252. "This supersaturation of dissolved N2O was obviously resulting from a net in-situ production of N2O in the water." This seems a reasonable interpretation based on the increase in N2O concentration in Figure 3; however, N2O saturation will also be affected by warming of the water from 16 to 24°C during the experiment. For example, N2O supersaturation could increase if the mesocosm water was ventilating at a slower rate than it was warming; this can be tested using the N2O gas exchange rate in Line 297.

To us it is unclear what meant by the statement of the reviewer. In line 252 ff. (of the original manuscript) we already discussed the apparent increase in $N_2O$ saturation, which resulted from the increase in water temperature and salinity over time. We agree that a reduction of ventilation (i.e. air sea gas exchange) towards the end of the study may have additionally resulted in an increase of $N_2O$ saturation. However, windspeed data towards the end of the study are not available due to technical problems. Therefore, we cannot test this assumption using the $N_2O$ gas exchange rate. We now added a sentence to the discussion part to clarify this issue (line 320 ff).

Line 282. "The time-series of N2O concentrations in both the SML and ULW showed no temporal trends". There is a suggestion in Figure 3 that N2O concentration in the ULW decreased during the C. closterium bloom relative to the rest of the experiment, and it would be interesting to examine this statistically.

We thank the reviewer for this comment. The apparent decrease in ULW $N_2O$ concentrations during the C. closterium bloom suggested by Figure 3 was now explicitly tested statistically. A generalized additive mixed model (GAMM) including a smooth term for time did not detect a significant temporal trend in $N_2O$ concentrations (smooth term for time: p = 0.27), indicating no systematic change over the course of the experiment. This result holds when accounting for irregular sampling frequency and short-term variability. In addition, paired comparisons between SML and ULW at individual sampling events showed no consistent or significant differences during the bloom phase. Taken together, these analyses do not support a statistically robust decrease in ULW $N_2O$ concentrations during the bloom period. We

therefore retain the statement that no temporal trend was detected and refrained from further interpretation, as attributing the visually perceived pattern to bloom-related processes would be speculative.

Conclusions

Line 312. "Consequently, a significant enrichment of N2O in the SML results in an enhanced N2O release to the atmosphere." No evidence for this is presented so this needs to be rewritten.

We appreciate this suggestion and rephrased the sentence to tone down the statement (lines 386 to 387).

Line 327. "we plea to develop a sampling method which minimizes the loss of dissolved trace gases…from the SML". SML methods are compared for non-gaseous substances in the Cunliffe & Wurl (2014) Best Practise, and there have been comparisons of SML techniques for dissolved gases largely focussed on DMS (Yang et al 2001). For example, Saint-Macary et al (2023) show that the glass plate measures lower DMS enrichment in the SML than a permeable tubing technique which could support the interpretation here and in the opening paragraph of the Discussion.

We thank the reviewer for the suggestion and now added the references and a brief discussion on the matter to the manuscript (lines 301 ff.). Additionally, we rephrased the sentence in line 327 (original manuscript) to highlight the need of refined methods to sample the SML for gases, while at the same time acknowledging that there have been scientific publications addressing this issue before.

**Minor**

Abstract

"Jade Bay (southern North Sea) water" – specify if this is coastal water and also location (Germany)

We thank the reviewer for the suggestion and included the information into the abstract.

Introduction

Line 52 "release of N2O FROM the water side"

Changed accordingly.

Line 55-56 "with the inherent problems of gas loss with the usually applied SML sampling methods (i.e. the glass plate and related methods)." Provide more information & references here

Thank you for pointing out. We added more information and added references to justify the statement.

Methods

Line 72 How do small pumps reduce biofilm formation?

We agree a biofilm formation cannot be prevented by small pumps and intermediate mixing, at most it can support a reduction in biofilm forming by reducing particle settlement. We therefore deleted this part from the sentence.

Fig 3. The y-axis scale for N2O concentration is too large; it can be reduced by half so that the variability in the data is more evident. Its also unclear why chlorophyll is shown in Fig 3 when theres no corresponding trend in N2O, and chlorophyll would be better combined in Figure 2 with nitrate and nitrite.

We thank the reviewer for the detailed comments on Figure 3. Regarding the y-axis scaling of $N_2O$ concentrations, the current scale was chosen deliberately. Reducing the y-axis range would visually amplify short-term variability and may misleadingly suggest the presence of a temporal trend. However, statistical analyses (GAMM and paired comparisons) clearly indicate that no significant temporal trend in $N_2O$ concentrations occurred over the course of the experiment. We therefore opted for a conservative scaling to avoid overinterpretation of visually apparent fluctuations that are not statistically supported. With respect to the inclusion of chlorophyll in Figure 3, this was done intentionally to directly relate phytoplankton bloom dynamics to $N_2O$ concentrations over time. One of the central objectives of this study was to assess whether phytoplankton blooms exert a detectable effect on $N_2O$ concentrations in the SML and ULW. The absence of a corresponding $N_2O$ response despite pronounced chlorophyll peaks is therefore a key result rather than a redundancy. Displaying chlorophyll alongside $N_2O$ explicitly illustrates this decoupling and supports the conclusion that, under the conditions of this experiment, phytoplankton blooms did not induce measurable changes in $N_2O$ concentrations. For these reasons, we consider the current structure of Figure 3 essential for conveying one of the main findings of the study. However, we have revised the figure caption to more explicitly guide interpretation and to clarify that no statistically significant temporal relationship between chlorophyll and $N_2O$ concentrations was detected.

Line 268 "*C. closteriuma*" typo

Changed accordingly.

Line 295 "Rough estimates" - honest, but perhaps not the best terminology in a publication

We thank the reviewer for pointing out this issue and changed the wording to "Approximate estimates".

Line 307. The last sentence finishes abruptly - "because sources and the…."

We apologize for the missing information, there must have been a cut-off during our own editing process. The sentence is nor complete again and reads as follows: "This could have led to the establishment of a steady state during the day time which persisted during night time because sources and the sink were not active during the night.

References

Saint-Macary, A.D., Marriner, A., Barthelmeß, T., Deppeler, S., Safi, K., Costa Santana, R., Harvey, M. and Law, C.S., 2023. Dimethyl sulfide cycling in the sea surface microlayer in the southwestern Pacific–Part 1: Enrichment potential determined using a novel sampler. *Ocean Science*, *19*(1), pp.1-15.

Yang, G.P., Watanabe, S. and Tsunogai, S., 2001. Distribution and cycling of dimethylsulfide in surface microlayer and subsurface seawater. Marine Chemistry, 76(3), pp.137-153.